

# Phase state and viscosity of secondary organic aerosols over China simulated by WRF-Chem

Zhiqiang Zhang[1,2], Ying Li[1], Haiyan Ran[1,2], Junling An[1], Yu Qu[1], Wei Zhou[1], Weiqi Xu[1], Weiwei Hu[3], Hongbin Xie[4], Zifa Wang[1], Yele Sun[1], Manabu Shiraiwa[5]

[1]State Key Laboratory of Atmospheric Boundary Layer Physics and Atmospheric Chemistry, Institute of Atmospheric Physics, Chinese Academy of Sciences, Beijing 100029, China

[2]College of Earth and Planetary Sciences, University of Chinese Academy of Sciences, Beijing 100049, China

[3]State Key Laboratory of Organic Geochemistry, Guangzhou Institute of Geochemistry, Chinese Academy of Sciences, Guangzhou 510640, China

[4]Key Laboratory of Industrial Ecology and Environmental Engineering (Ministry of Education), School of Environmental Science and Technology, Dalian University of Technology, Dalian 116024, China

[5]Department of Chemistry, University of California, Irvine, CA 92697-2025, USA

*Correspondence to*: Ying Li (liying-iap@mail.iap.ac.cn)

**Abstract.** Secondary organic aerosols (SOA) can exist in liquid, semi-solid or amorphous solid states, which are rarely accounted for in current chemical transport models (CTMs). Missing the information of SOA phase state and viscosity in CTMs impedes accurate representation of SOA formation and evolution, affecting the predictions of aerosol effects on air quality and climate. We have previously developed a method to estimate the glass transition temperature ($T_g$) of an organic compound based on volatility. In this study, we apply this method to predict the phase state and viscosity of SOA particles over China in summer of 2018 using the Weather Research and Forecasting model coupled to Chemistry (WRF-Chem). This is the first time that spatial distributions of the SOA phase state over China are investigated by a regional CTM. Simulations show that $T_g$ values of dry SOA range from ~287 K to 305 K, with higher values in the northwestern China where SOA particles have larger mass fractions of low volatility compounds. Considering water uptake by SOA particles, the SOA viscosity also shows a prominent geospatial gradient that highly viscous or solid SOA particles are mainly found in the northwestern China. The lowest and highest SOA viscosity values both occur over the Qinghai-Tibet Plateau that the solid phase state is predicted over dry and high-altitude areas and the liquid phase state is predicted mainly in the south of the plateau with high relative humidity during the summer monsoon season. The characteristic mixing timescale of organic molecules in 200 nm SOA particles is calculated based on the simulated particle viscosity and the bulk diffusion coefficient of organic molecules. Calculations show that during the simulated period the percent time of the mixing timescale longer than 1 h is > 70 % at the surface and at 500 hPa in most areas of the northern China, indicating that kinetic partitioning considering the bulk diffusion in viscous particles may be required for more accurate prediction of SOA mass concentrations and size distributions over these areas. Sensitivity simulations show that including the formation of extremely low-volatile organic compounds, the percent time that a SOA particle is in the liquid phase state decreases by up to 12 % in the





southeastern China during the simulated period. With an assumption that the organic and inorganic compounds are always
internally mixed in one phase, we show that the water absorbed by inorganic species can significantly lower the simulated
viscosity over the southeastern China. This indicates that constraining the uncertainties in simulated SOA volatility
distributions and accurately predicting the occurrence of phase separation would improve prediction of viscosity in
multicomponent particles in southeastern China.
**1 Introduction**
Secondary organic aerosols (SOA) are major components of atmospheric fine particles, impacting air quality, climate and
public health (Jimenez et al., 2009; Pöschl and Shiraiwa, 2015). The formation and evolution of SOA involve both chemical
reactions and mass transport in the gas and particle phases (Ziemann and Atkinson, 2012). This complexity makes accurate
representation of SOA evolution in chemical transport models (CTMs) challenging, leading to a large uncertainty in
evaluating SOA impacts on air quality and climate (Kanakidou et al., 2005; Shrivastava et al., 2017).

Current CTMs usually assume that SOA particles are homogeneous and well-mixed liquids, with rapid establishment of

gas–particle equilibrium applied in simulations of SOA formation and partitioning (Pankow, 1994; Donahue et al., 2006). It
has been shown that SOA can exist in liquid (low dynamic viscosity $\eta$, $\eta < 10^2$ Pa s), semi-solid ($10^2$ Pa s $\leq \eta \leq 10^{12}$ Pa s) or
solid (amorphous or glassy solid; $\eta > 10^{12}$ Pa s) states, depending on particle chemical composition and atmospheric
conditions, such as ambient temperature ($T$) and relative humidity (RH) (Koop et al., 2011; Reid et al., 2018). Viscosities can
be converted to bulk diffusion coefficients via the Stokes-Einstein equation (Einstein, 1905; Seinfeld and Pandis, 2016) or
the fractional Stokes-Einstein equation (Price et al., 2016; Evoy et al., 2019; Evoy et al., 2020). The phase state, viscosity,
and bulk diffusivity of SOA are important in many aerosol processes. The semi-solid or solid phase state can prolong the
equilibration timescales in the gas-particle partitioning, indicating a need of considering kinetic limitations in the SOA
partitioning into highly viscous particles (Shiraiwa and Seinfeld, 2012; Roldin et al., 2014; Zaveri et al., 2014; Li and
Shiraiwa, 2019). The viscosity of SOA can impact the rates of heterogeneous and multiphase reactions (Marshall et al., 2018;
Zhang et al., 2019a), photochemistry (Liu et al., 2018; Dalton and Nizkorodov, 2021; Baboomian et al., 2022), and the
uptake of gaseous pollutants (e.g., $O_3$, OH, $N_2O_5$, $NO_2$, $NH_3$, and $SO_2$) and water vapor (Abbatt et al., 2012; Kuwata and
Martin, 2012; Preston and Zuend, 2022), with implications for accurate predictions of atmospheric chemical composition
(Reid et al., 2018). The SOA phase state also affects particle size distribution evolution (Shiraiwa et al., 2013; Zaveri et al.,
2018) and ice nucleation pathways (Knopf and Alpert, 2023).

Accurate predictions of the viscosity need the information of molecular structures and functional groups (Song et al.,

2016; Rothfuss and Petters, 2017; Gervasi et al., 2020; Galeazzo and Shiraiwa, 2022); however, molecular specificity is
often difficult to be obtained in ambient measurements. Currently there are only a few methods developed to predict the



phase state of ambient SOA particles, and successfully be implemented in CTMs. Li, Shiraiwa and coauthors first developed
a parameterization predicting the glass transition temperature ($T_\mathrm{g}$) based on the molar mass ($M$) and the atomic O/C ratio for
carbon-hydrogen (CH) and carbon-hydrogen-oxygen (CHO) compounds with their molar mass less than 450 g mol$^{-1}$
(Shiraiwa et al., 2017). $T_\mathrm{g}$ characterizes the temperature at which a phase transition between amorphous solid and semi-solid
states occurs (Koop et al., 2011). When the ambient $T$ is higher than $T_\mathrm{g}$, a SOA particle is in a semi-sold or liquid phase state;
otherwise, it behaves as an amorphous solid. This parameterization has been successfully coupled into CTMs simulating the
SOA phase state over the globe (Shiraiwa et al., 2017) or the U.S. (Schmedding et al., 2020; Li et al., 2021b), showing that
semi-solid and amorphous solid phase states frequently occurred in ambient SOA particles over dry lands or in the upper
troposphere. Further parameterizations were developed to predict $T_\mathrm{g}$ as a function of the saturation mass concentration ($C^0$)
and the O/C ratio of organic compounds, or as a function of $C^0$ solely, which indirectly included the molecular structure
effect on viscosity estimations (Li et al., 2020). This parameterization was added into the Weather Research and Forecasting
Model coupled to chemistry (WRF-Chem) (Grell et al., 2005; Fast et al., 2006), and the simulations showed that the
viscosity of SOA particles could be reasonably predicted during the dry-to-wet transition season in the Amazon rainforest
(Rasool et al., 2021). Instead of predicting $T_\mathrm{g}$, Maclean et al. developed parameterizations for viscosity as a function of $T$ and
RH based on measured viscosity data of laboratory SOA, and applied the viscosity parameterizations in CTMs to predict the
mixing timescales of organic molecules and water molecules within SOA particles (Maclean et al., 2017; Maclean et al.,

2021).

Investigations in the particle phase state over China are currently focused on field observations and laboratory

experiments. Bounce factor measurements showed that submicrometer particles can be semi-solid in clear days and liquid in
hazy days in Beijing, China (Liu et al., 2017). The phase state of PM$_{2.5}$ (particulate matter with an aerodynamic diameter $\leq$
2.5 μm) was found to be mostly semisolid to solid in winter Beijing based on the measurements using optical microscopy
combined with the poke-and-flow technique (Song et al., 2022). The RH-dependent viscosity of the proxies of actual
ambient particles in Beijing was also investigated based on dual optical tweezers (Tong et al., 2022). The phase state of
submicrometer particles in Beijing was retrieved from a polarization lidar that has the potential to infer the vertical profiles
of phase state (Tan et al., 2020). The phase state of traffic-related secondary aerosols in Beijing may have a distinguished
diurnal variation (Meng et al., 2021). The biomass burning aerosols, collected near a farmland in Yangtze River Delta, China,
were found to exist in the non-solid phase state at relatively dry conditions (Liu et al., 2021b).

These measurements indicate that the particle phase state over China is highly variable under different atmospheric

conditions. It is important to know the spatial distributions and time variations of the SOA phase state and viscosity in
multicomponent particles to better quantify the aerosol effects on air quality, which, however, has not been investigated over
China with air quality models on a regional scale. Here we use the WRF-Chem model simulated SOA volatility distributions
to estimate the glass transition temperature and viscosity of SOA particles over China based on the parameterizations



developed in Li et al. (2020). We further calculate the diffusion coefficients and mixing timescales of organic molecules
within SOA, which has implications in how to properly treat the SOA partitioning (instantaneous equilibrium vs. kinetic
partitioning) in CTMs. As volatility and viscosity are closely related, we conduct a sensitivity calculation to evaluate the
effects of the simulated SOA volatility distributions on viscosity estimations. We also conduct a sensitivity calculation to
investigate how the water absorbed by inorganic components in $PM_{2.5}$ affects viscosity estimations, which has implications
in predicting the viscosity of internally mixed ambient particles.
**2 Methods**
**2.1 WRF-Chem model configuration**
We use the WRF-Chem model version 3.7.1 (Grell et al., 2005; Fast et al., 2006) and simulate the period from 20 May to 23
June 2018 with a spin-up period of 7 days (May 13 – 19). The meteorological initial and boundary conditions are from the
National Centers for Environmental Prediction (NCEP) Global Forecast System (GFS) final (FNL) reanalysis data. The
outputs of a global chemical transport model MOZART-4 (Emmons et al., 2010) provide initial and boundary conditions of
chemical species over the outer domain. Anthropogenic emissions are from the MIX 2010 inventory for Asia (Li et al., 2017)
and the MEIC 2016 inventory for China (http://meicmodel.org.cn) (Zheng et al., 2018). Biogenic emissions are calculated
from the Model of Emissions of Gases and Aerosols from Nature (MEGAN2.1) (Guenther et al., 2012). We set up two
domains (Fig. S1) with the horizontal resolutions of 81 km and 27 km, respectively, and 18 vertical layers are applied from
the surface up to 100 hPa.
The utilized physical and chemical schemes are given in Table S1. We use the MOZART-4 mechanism (Emmons et al.,
2010) for the gas-phase chemistry. The MOSAIC (Model for Simulating Aerosol Interactions and Chemistry) aerosol module
(Zaveri et al., 2008) is applied for the aerosol chemistry and we represent aerosol particles with 4-size sections having dry
diameters ranging from 39 nm to 10 μm (Knote et al., 2015). SOA formation is treated with the 1-D volatility basis set (VBS)
approach (Donahue et al., 2006) which has been implemented into the MOSAIC aerosol module (Lane et al., 2008b;
Ahmadov et al., 2012). Five volatility bins are considered (effective saturation mass concentrations $C^*$ of $10^{-4}$, 1, 10, 100,
and 1000 μg m$^{-3}$ at 298 K) in the official version 3.7.1 of the WRF-Chem model, with the enthalpy of vaporization ($\Delta H_{vap}$)
values of 40, 131, 120, 109, and 98 kJ mol$^{-1}$ used in each volatility bin (Knote et al., 2015). We follow Knote et al. (2015)
with SOA mass yields adopted for four volatility bins (1, 10, 100, and 1000 μg m$^{-3}$). Further gas-phase aging is simulated
through OH oxidation of SOA vapors with a fixed rate of $1.0 \times 10^{-11}$ cm$^3$ molec$^{-1}$ s$^{-1}$, with products shifted down one
volatility bin (Murphy and Pandis, 2009), e.g., the condensable vapors with $C^*$ of 1 μg m$^{-3}$ react with OH forming surrogate
species in the lowest volatility bin ($C^*$ of $10^{-4}$ μg m$^{-3}$). The partitioning of organic compounds between the gas and particle
phases is simulated based on Pankow (1994) which is implemented in MOSAIC (Shrivastava et al., 2011). We apply glass



transition temperature and viscosity calculations to WRF-Chem model output for traditional SOA formed from the oxidation
of volatile organic compounds including alkanes, alkenes, aromatics, isoprene, and monoterpenes (Lane et al., 2008a).

**2.2 Glass transition temperature and viscosity calculations**

The glass-transition temperature of SOA products in each volatility bin at dry conditions ($T_{g,i}$) is calculated as a function of
the saturation mass concentration at 298 K ($C^0$) using the parameterization (Eq. 1) developed in our previous study (Li et al.,
2020). We assume ideal thermodynamic mixing that $C^0$ is equal to $C^*$, which is often applied in the VBS (Donahue et al.,

2011).

$$T_{g,i} = 288.70 - 15.33 \times log_{10}(C^0) - 0.33 \times [log_{10}(C^0)]^2 \qquad (1)$$
The $T_g$ of mixtures of dry SOA compounds ($T_{g,org}$) is calculated by the Gordon−Taylor equation (Gordon and Taylor,
1952), with the Gordon−Taylor constant ($k_{GT}$) assumed to be 1 (Dette et al., 2014):
$$T_{g,org} = \sum_i \omega_i\, T_{g,i} \qquad (2)$$
where $\omega_i$ is the mass fraction of SOA products in each volatility bin simulated by the VBS module in WRF-Chem.
The particle phase state depends strongly on water content in particles, as water can act as a plasticizer to decrease
viscosity (Mikhailov et al., 2009; Koop et al., 2011). The mass concentration of water absorbed by SOA particles under
humid conditions is estimated using the effective hygroscopicity parameter ($\kappa$) (Petters and Kreidenweis, 2007) as:
$$m_{H_2O} = \left(\frac{a_w}{1 - a_w}\right)\frac{\kappa \rho_w m_{SOA}}{\rho_{SOA}} \qquad (3)$$
where $a_w$ is water activity calculated as $a_w$ = RH/100 and $\rho_w$ is the density of water. $m_{SOA}$ is the simulated total mass
concentrations of traditional SOA. The density of SOA particles ($\rho_{SOA}$) is assumed to be 1.5 g cm$^{-3}$ (Knote et al., 2015). $\kappa$ is
assumed to be 0.1 based on previous studies (Gunthe et al., 2009; Duplissy et al., 2011; Wu et al., 2013) and consistent with
the value used in our previous global SOA phase state simulations (Shiraiwa et al., 2017).
$T_g$ of organic-water mixtures is also calculated by the Gordon−Taylor equation (Eq. 4) with $k_{GT}$ suggested to be 2.5
(Koop et al., 2011).
$$T_g(\omega_{org}) = \frac{(1 - \omega_{org})T_{g,w} + \frac{1}{k_{GT}}\omega_{org}T_{g,org}}{(1 - \omega_{org}) + \frac{1}{k_{GT}}\omega_{org}} \qquad (4)$$
where $\omega_{org}$ is the mass fraction of the simulated SOA species in organic-water mixtures. The glass transition temperature of
pure water ($T_{g,w}$) is 136 K (Kohl et al., 2005). Based on $T_g(\omega_{org})$, viscosity can be calculated with the
Vogel−Tammann−Fulcher (VTF) equation (Angell, 1991): $\eta = \eta_\infty e^{\frac{T_0 D}{T - T_0}}$, where $\eta_\infty$ is the viscosity at infinite temperature
($10^{-5}$ Pa s, Angell (1991)). $D$ is the fragility parameter which is adopted to be 10 based on our previous study in DeRieux and



Li et al. (2018). $T_0$ is the Vogel temperature calculated as $T_0 = \frac{39.17 T_g(\omega_{org})}{D+39.17}$. We further calculate the bulk diffusion
coefficient ($D_b$) of organic molecules with a radius of 0.4 nm (Maclean et al., 2021) and water molecules in SOA particles
based on predicted viscosity and the fractional Stokes−Einstein equation (Price et al., 2016; Evoy et al., 2019; Evoy et al.,
2020), which is detailed in the Supplement. The mixing timescales of molecules ($\tau_{mix}$) within SOA particles is calculated as
$\tau_{mix} = d_p^2/(4\pi^2 D_b)$ (Seinfeld and Pandis, 2016), where $d_p$ is the particle diameter. The $d_p$ is assumed to be 200 nm
(Maclean et al., 2021) when we calculate $\tau_{mix}$.
**2.3 Sensitivity simulations**
Table 1 lists all the performed simulations. In the base case, we update the $C^*$ in the lowest volatility bin from $10^{-4}$ µg m$^{-3}$ in
the official WRF-Chem v3.7.1 to 0.1 µg m$^{-3}$ based on the ambient volatility observations (referring to Section 2.4 and Fig. 1),
and calculate the $\Delta H_{vap}$ in the lowest volatility bin using the semi-empirical parameterization in Epstein et al. (2010), leading
to a value of 142 kJ mol$^{-1}$. To evaluate the effects of simulated SOA volatility distributions on phase state estimations, we
conduct a simulation (sensitivity case A) following the default setting in the model assuming that the lowest $C^*$ is $10^{-4}$ µg m$^{-}$
$^3$ at 298 K, with $\Delta H_{vap}$ of 40 kJ mol$^{-1}$ (Knote et al., 2015). A smaller $\Delta H_{vap}$ indicates less dependence of volatility on
temperature variations. In the sensitivity case B, we increase the simulated RH by a factor of 10 % as we find that the
simulated RH values are smaller than the observations (Section 3.1). In the base case and sensitivity cases A and B, we
predict $T_g$ for SOA-water mixtures accounting for the SOA-influenced water uptake solely, assuming that SOA particles are
externally mixed with inorganic compounds such as sulfate and nitrate. In the sensitivity case C, we assume that the organic
and inorganic compounds are always internally mixed in one phase and include the water absorbed by inorganic compounds
in viscosity calculations. The water associated with inorganics is calculated by the MOSAIC module coupled in WRF-Chem.
**2.4 Observation**
The observation data measured at an urban site in the Institute of Atmospheric Physics (IAP), Chinese Academy of Sciences
(39°58′28″ N, 116°22′16″ E) in Beijing (Fig. S1) are used to compare with the simulation results of the WRF-Chem model.
The aerosol volatility was measured from 20 May to 23 June in 2018, using a thermodenuder coupled with an Aerodyne
high-resolution aerosol mass spectrometer (Xu et al., 2019). The volatility distributions of oxygenated organic aerosols
(OOA) resolved from positive matrix factorization (PMF) were estimated using a dynamic mass transfer model (Riipinen et
al., 2010). The volatility of OOA was found to be distributed in six logarithmically spaced $C^*$ bins including 0.001, 0.01, 0.1,
1, 10, and 100 µg m$^{-3}$, based on the best fits between the measured and predicted thermograms using the methods in Karnezi
et al. (2014). Chemical species including organics (Org), sulfate (SO$_4^{2-}$), nitrate (NO$_3^-$), and ammonium (NH$_4^+$) in PM$_{2.5}$
were measured using an Aerodyne time-of-flight aerosol chemical speciation monitor (Fröhlich et al., 2013) equipped with a
capture vaporizer and PM$_{2.5}$ lens, with the details described in Li et al. (2023). The OOA factor was identified with the PMF



analysis. We obtain the mass concentrations of PM$_{2.5}$ from the Olympic Center observation site (http://zx.bjmemc.com.cn)
which is ~4 km from the IAP site (Fig. S1). Meteorological parameters including RH and $T$ are from the Beijing
meteorological tower at the IAP site.

**3 Results**


**3.1 Simulations in Beijing and the comparison with observations**


The comprehensive model evaluations were conducted in our previous studies, showing that the WRF-Chem model
reasonably captured the magnitudes and spatial distributions of concentrations of major air pollutants over China (Li et al.,
2011; Li et al., 2014; Li et al., 2015; Qu et al., 2019; Zhang et al., 2022). Here we focus on the comparison of simulations
and observations at the IAP site during 20 May – 23 June 2018 when the observed volatility distributions are available (Xu et
al., 2019).
Figure 1 shows the average volatility distributions of observed OOA and simulated SOA at the IAP site. The $C^*$ of OOA
spans from 0.001 µg m$^{-3}$ to 100 µg m$^{-3}$, with an average value of 1.16 µg m$^{-3}$. The semi-volatile organic compounds (SVOC;
$0.3 < C^0 < 300$ µg m$^{-3}$) and the low-volatile organic compounds (LVOC; $3×10^{-4} < C^0 < 0.3$ µg m$^{-3}$) (Donahue et al., 2012)
contribute 66.3 % and 33.7 % to OOA concentrations, respectively (Xu et al., 2019). The $T_{g,org}$ estimated from the observed
OOA volatility distributions is 286.7 K. Figure 1a shows the simulated volatility distributions of SOA with five $C^*$ bins set to
be 0.1, 1, 10, 100, and 1000 µg m$^{-3}$ at 298 K, and $\Delta H_{vap}$ of 142, 131, 120, 109, and 98 kJ mol$^{-1}$ used in the five $C^*$ bins,
respectively (base case in Table 1). In this base case simulation, the SOA consists of 64.5 % SVOC and 35.3 % LVOC, and
most of the SVOC species are located in the $C^*$ bin of 1 µg m$^{-3}$. The simulated SOA in Fig. 1a has an average $C^*$ of 0.64 µg
m$^{-3}$ and $T_{g,org}$ of 291.5 K, close to the values estimated from the volatility distributions of OOA. Figure 1b shows the
simulated volatility distributions of SOA with the lowest $C^*$ bin set to be 0.0001 µg m$^{-3}$ at 298 K with $\Delta H_{vap}$ of 40 kJ mol$^{-1}$,
following the default option in the official WRF-Chem model 3.7.1 (Knote et al., 2015). In this sensitivity simulation (case A
in Table 1), the SOA consists of 40.4 % extremely low-volatile organic compounds (ELVOC; $C^0 < 3×10^{-4}$ µg m$^{-3}$), which are
not determined in the observed OOA, leading to a much lower average $C^*$ (0.03 µg m$^{-3}$) and a higher $T_{g,org}$ (309.0 K)
compared to the observations. In the following we estimate the $T_{g,org}$ and viscosity of SOA using the simulated volatility
distributions in the base case with the lowest $C^*$ bin set as 0.1 µg m$^{-3}$ at 298 K. The impacts of volatility distributions with
the incorporation of ELVOC (Fig. 1b) on viscosity estimations are evaluated in section 3.3.
Figure 2 shows that the model relatively well reproduces the observed hourly variations of RH, $T$, mass concentrations
of PM$_{2.5}$ and its major inorganic components (Figs. 2a – f), with the index of agreement (IOA, defined in the supplement)
varied from ~0.70 for inorganic components to a higher value of 0.93 for $T$ (Table S2). The simulated values of RH are
constantly lower than the observations, with the mean bias (MB) being –10.97 % (Table S2). The underestimation of RH



observations was also found at other meteorological sites in the North China Plain in our previous studies (Qu et al., 2019;
Zhang et al., 2022), which would affect the SOA viscosity estimations. The effects of RH on the viscosity estimations are
evaluated in section 3.3. Figure 2g shows that the model could generally reproduce the observed temporal variations of OOA
concentrations, but largely underestimates the observation peaks (MB = –5.88 μg m$^{-3}$, the normalized mean bias NMB = –
53.28 %, Table S2). Incorporation of the SOA formed from intermediate-volatile organic compounds (IVOCs) (Miao et al.,
2021; Chang et al., 2022) would increase the simulated SOA concentrations, which is beyond the scope of this study and will
be considered in our future work. The simulated SOA mean concentration is 5.15 μg m$^{-3}$. Although it is lower than the
observed value of 11.03 μg m$^{-3}$, this difference in the simulated and observed SOA concentrations would not affect the
viscosity predictions significantly. The SOA viscosity has a much closer relation with the volatility rather than its mass
loadings (Li et al., 2020; Champion et al., 2019). In our previous study we estimated $T_{g,org}$ and viscosities at different OA
mass loadings varied from 1 to 1000 μg m$^{-3}$, showing that the simulated viscosities were very similar, particularly when RH
was higher than 50 % (Derieux et al., 2018).
As the WRF-Chem model underestimates the observed RH at the IAP site, we calculate the SOA viscosity using the
simulated and observed RH, respectively. Figure 2h shows that the viscosities calculated at the two conditions are similar at
most times during the simulated period, ranging mainly from ~$10^2$ Pa s to $10^{10}$ Pa s, with a median value of ~$10^7$ Pa s,
indicating that a semi-solid phase state frequently occurs. The underestimations of the observed RH by WRF-Chem mainly
impact the phase state estimations at relatively high RH. For example, SOA particles occur as liquid when the observed RH
is higher than ~75 %; however, they remain in a semi-solid phase state at the simulated RH. The bulk diffusion coefficients
($D_b$) of organic molecules range from $10^{-18}$ to $10^{-11}$ cm$^2$ s$^{-1}$ at the simulated RH (Fig. 2i), leading to the mixing timescales
within 200 nm SOA particles being seconds to years, with 61 % of the time > 1 hour (Fig. S2).
The vertical profiles of SOA viscosity exhibit diurnal variations. Figure 3a shows the median diurnal and vertical
profiles of predicted SOA viscosity at the IAP site. The SOA particles remain highly viscous (~$10^7 – 10^8$ Pa s) at the surface,
with a higher viscosity occurring from late afternoon to early evening, during which the RH is less than 20 %, lower than the
rest time of the day (Fig. S3). The SOA particles become more viscous at higher altitudes than the surface and adopt the
phase transition from a semi-solid phase to a solid phase at ~4 km at the IAP site. The predicted altitude with the phase
transition is ~2 km higher than our previous global model prediction for the region of East China which was an average of
five years' simulations (Shiraiwa et al., 2017). Tan et al. (2020) inferred the phase state of submicrometer particles in Beijing
from the surface to an altitude of ~1 km using a polarization lidar and found that the particle phase state exhibits a vertical
variation. Further observations of SOA viscosity at high altitudes are needed to better understand the viscosity vertical
profiles and validate our predictions. Figure 3b shows that the mixing timescales for organic molecules within 200 nm SOA
particles are approximately 1 hour at the surface, and longer than ~10 hours at altitudes higher than 1 km, indicating that
kinetic limitations in the gas-particle partitioning may be required to accurately predict SOA mass concentrations in summer



Beijing, particularly in the upper planetary boundary layer and the free troposphere.
**3.2 Simulated glass transition temperature and viscosity of SOA particles over China**
The glass transition temperature of the dry organic phase ($T_{g,org}$) shows a geospatial gradient over China. Figure 4a shows the
median surface values of $T_{g,org}$ calculated in the base simulation (Table 1). $T_{g,org}$ ranges from ~287 – 305 K over most areas of
China, with lower values occurring mainly over the southeast and higher values over the northwest. The $T_{g,org}$ range
simulated by the WRF-Chem model is consistent with our previous global simulations of $T_{g,org}$ that varied from ~285 K to
310 K at the surface over China (Shiraiwa et al., 2017). The geospatial variation in $T_{g,org}$ is related to the simulated SOA
volatility distributions. Figure 4b shows the mass fractions of SOA species distributed in the lowest volatility bin (SOAX
with $C^* = 0.1 \ \mu g \ m^{-3}$ at 298 K). The mass fractions of SOAX are mostly 20 – 35 % in the southeastern China, indicating that
the majority of the simulated SOA formed from VOCs is semi-volatile. In these areas, the simulated SOA mass
concentrations are higher than the other locations of China (Fig. S4) (Li et al., 2022), which is favorable for more SVOCs
partitioning into the particle phase, leading to relatively low values of $T_{g,org}$ (Fig. 4a). LVOCs are more frequently contained
in aged SOA particles in remote areas, e.g., some areas in the northwestern China where the SOA mass concentrations are
very low, resulted in higher $T_{g,org}$ values.
The relative humidity plays an important role in regulating SOA viscosity (Koop et al., 2011). Considering the water
uptake by SOA particles in the phase state estimations, the predicted geospatial patterns in the viscosity (Fig. 5a) and RH
(Fig. S4) are very similar with each other, particularly in southern and northeastern China. SOA particles are predicted to
mainly be liquid or with a low viscosity (< $10^4$ Pa s) in the southeast. Figure 5b shows the frequency of liquid phase state,
which is calculated as the percent time that an organic aerosol particle is in the liquid phase state during the simulated period.
The frequency of liquid particles varies from ~ 30 % to 70 % in the southeastern China. The lowest viscosity with the highest
frequency of liquid particles occurs over the southern Tibetan Plateau where RH is very high (Fig. S4), which is contributed
by summer monsoons and regional moisture recycling (Dong et al., 2016). The SOA particles in the central and northeastern
China are predicted to be semi-solid, with the viscosity varied from $10^5$ to $10^8$ Pa s (Fig. 5a). Highly viscous ($\eta > 10^8$ Pa s) or
solid SOA particles are mainly found in the northwest, particularly over the northern Tibetan Plateau where the ambient
temperatures are lower than other areas of China (Fig. S4). The frequency of liquid SOA particles in most areas with the
latitude higher than 30°N is less than 20 % (Fig. 5b).
The simulated geospatial pattern in SOA viscosity over China agrees with previous global simulations and ambient
measurements. Our previous global simulations predicted a lower viscosity ($\eta < 10^3$ Pa s) in SOA particles in southeastern
China and a higher viscosity ($\eta > 10^8$ Pa s) in northwestern China (Shiraiwa et al., 2017; Li et al., 2020), similar to the
WRF-Chem simulations in this study. Interestingly, the occurrence of liquid particles over the southern Tibetan Plateau in
summer simulated by the WRF-Chem was not found in our previous global predictions, which was an average of five years'



simulations. The semi-solid phase state of SOA particles simulated in Beijing is consistent with both particle bounce
measurements (Liu et al., 2017) and the $PM_{2.5}$ phase state determined by the poke-and-flow technique (Song et al., 2022).
The simulated viscosity of SOA particles is 0.15 Pa s in Shenzhen, a coastal urban city in southeastern China, which also
agrees with the findings in the previous bounce measurements indicating that the submicron particles in Shenzhen are in the
liquid state (Liu et al., 2019).

Figure 5c shows the median values of viscosity as a function of RH calculated for selected regions in the northern China,

southern China, northern Qinghai-Tibet Plateau, and southern Qinghai-Tibet Plateau as specified by white boxes in Fig. 5a.
There is a strong inverse relationship between SOA viscosity and relative humidity with high RH (> ~ 60 %) as the dominant
factor determining the phase state of SOA particles. When RH is lower than ~ 60 %, the predicted viscosity is affected by
both RH and $T$. For example, the SOA particles occur mainly as solid over the Northern Qinghai-Tibet Plateau while occur
as semi-solid over the Northern China within similar RH ranges (20 % < RH < 60 %); the reason is that the ambient $T$ over
the Northern Qinghai-Tibet Plateau is much lower (~ 20 K lower) than the Northern China (Fig. S5a). When RH is relatively
low, the viscosity of SOA particles is also influenced by particle chemical composition, i.e., the SOA particles composed of
high mass fractions of low volatility compounds tend to have higher viscosity values (Fig. S5b). RH is the main factor
driving the diurnal variations of SOA viscosity in our simulations. Figure S6 shows that SOA particles have higher viscosity
in the daytime than the nighttime as RH in the daytime is lower than the nighttime (Fig. S7). Compared to the northern China,
the southern China exhibits stronger diurnal variations in SOA viscosity that SOA particles occur mainly as semi-solid in the
daytime and liquid in the nighttime. Highly viscous or solid SOA particles are found in the northern China during both
daytime and nighttime (Fig. S6).

The bulk diffusion coefficient is an important parameter determining the mass-transport and mixing rates, which can be

predicted by the particle viscosity through the fractional Stokes-Einstein relation (see the Method). The $D_b$ of organic
molecules is predicted to be > ~$10^{-10}$ cm$^2$ s$^{-1}$ in the southern China. The highest value is ~$10^{-5}$ cm$^2$ s$^{-1}$ occurring in liquid
SOA particles in the southern Tibetan Plateau (Fig. 6a). The $D_b$ of organic molecules within semisolid SOA particles is ~$10^{-18}$
$^{18}$ – $10^{-10}$ cm$^2$ s$^{-1}$ in the central and northeastern China, and lower than ~$10^{-18}$ cm$^2$ s$^{-1}$ in highly viscous and solid particles in
most areas of the northwestern China. Figure 6b shows the percent time that the mixing timescale of organic molecules in
200 nm particles is less than 1 h in the entire simulation period. The mixing timescale is found nearly always less than 1 h in
the southeastern region of the "Hu Huanyong Line". The "Hu Huanyong Line", proposed by the Chinese geographer
Huanyong Hu, divides China into two parts based on contrasting population densities (Hu, 1935), which was found also
useful characterizing the drought conditions, with the northwestern region much drier than the southeastern region (Zeng et
al., 2021). The mixing timescale of organic molecules in highly viscous or solid SOA particles in the northwest of the "Hu
Huanyong Line" is often longer than 1 h (the frequency > 70 %), indicating that in these areas kinetic limitations of bulk
diffusion should be considered in SOA partitioning. Compared to the diffusion coefficients of organic molecules, the $D_b$ of



water molecules in SOA particles at the surface is several orders of magnitude larger, with the values higher than $10^{-10}$ cm$^2$ s$^-$
$^1$ in the southeast, and as low as ~ $10^{-13}$ cm$^2$ s$^{-1}$ in the northwestern China (Fig. 6c). The mixing timescales of water
molecules in SOA particles with a diameter of 200 nm are of the order of milliseconds in the southeast and seconds in the
northwest of China (Fig. S8), indicating that the activation of cloud condensation nuclei would not be inhibited, in agreement
with our previous global simulations (Shiraiwa et al., 2017).
Figure 7 shows the simulated $T_{g,org}$ and the phase state of SOA particles, as well as the mixing timescale of organic
molecules in SOA particles at 500 hPa. The $T_{g,org}$ ranges from ~ 285 K – 295 K, lower than the $T_{g,org}$ simulated at the surface
(Fig. 7a). The reason is that the mass fractions of LVOCs (SOAX with $C^*$ of 0.1 μg m$^{-3}$ at 298 K) at 500 hPa (Fig. S9) are
smaller than the surface values (Fig. 4b). The low temperature at 500 hPa is favorable for SVOCs partitioned into the particle
phase, thus compared to the surface conditions, there is less semi-volatile vapors undergoing the further gas-phase aging
forming SOAX species. The percent time that an organic aerosol particle is in the liquid phase state (the frequency of liquid
SOA particles) at 500 hPa in the southeastern China is 20 – 35 % (Fig. 7b), which is ~ 20 % lower than the surface values
(Fig. 5b). In the northern China, the frequency of liquid SOA particles at 500 hPa is similar to the results at the surface,
which is related to the RH spatial patterns (Fig. S10). The mixing timescale of organic molecules in 200 nm SOA particles is
frequently longer than 1 h at 500 hPa, with the frequency > 70 % in the northern China and ~ 40 % in the southeastern China
(Fig. 7c). The $\tau_{mix}$ is relatively short (the frequency of $\tau_{mix} \leq 1$ h being ~ 80 %) in some areas of the southwestern China at
500 hPa, where relatively high RH could occur (Fig. S10) in the season of summer monsoon (Huang et al., 1998).
**3.3 Sensitivity simulations**
**3.3.1 Impacts of volatility distributions on phase state estimations**
The volatility and viscosity of organic aerosols are closely related (Rothfuss and Petters, 2017; Shiraiwa et al., 2017;
Champion et al., 2019; Zhang et al., 2019b; Li et al., 2020). In this section we conduct sensitivity simulations (case A, Table
1) to evaluate how the simulated volatility distributions affect the phase state estimations. The lowest $C^*$ bin in the base case
is 0.1 μg m$^{-3}$ at 298 K, with $\Delta H_{vap}$ of 142 kJ mol$^{-1}$, which does not incorporate ELVOC species at the room temperature. The
sensitivity simulation (case A) adopts the default setting in the official WRF-Chem model v3.7.1, assuming that the lowest
$C^*$ is 0.0001 μg m$^{-3}$ at 298 K, with $\Delta H_{vap}$ of 40 kJ mol$^{-1}$ (Knote et al., 2015). Figure 8 shows that including these ELVOCs at
298 K with a relatively small $\Delta H_{vap}$ mainly affect the $T_{g,org}$ simulated over remote areas, e.g., the northwestern China and the
marine areas, where the simulated $T_{g,org}$ is increased by 30 – 40 K (Fig. 8a). In other regions of China, the changes in $T_{g,org}$
are less than 25 K. Although consideration of these ELVOCs could affect the simulated $T_{g,org}$ at the surface obviously in
remote areas, i.e., the northwestern China, it does not impact the predicted frequency of the occurrence of a liquid phase state
(Fig. 8b), as in these dry areas the SOA particles are highly viscous. Including ELVOC formation at 298 K ($\Delta H_{vap}$ of 40 kJ
mol$^{-1}$) mainly affects the phase state estimations in areas with a moderate humidity. For example, in some areas of the





southeastern China with ~ 70 % RH, the SOA particles are predicted to be more viscous, with the frequency of a liquid phase
state decreasing by up to 12 % (Fig. 8b). These results indicate that the SOA phase state estimations in the base and
sensitivity case A are generally in agreement in the simulated episode in this study. Ambient measurements of organic
aerosol volatility distributions are still sparse over China. A recent field study showed that the ELVOCs contributed more
than half to the OA mass observed at a regional background site near the Bohai Sea (Feng et al., 2023), which resulted in an
estimated viscosity much higher than our WRF-Chem simulations. More field volatility distribution measurements should be
conducted over China to further evaluate the effects of ELVOCs and how to choose reasonable values of the enthalpy of
vaporization that would affect phase state estimations.
**3.3.2 Impacts of RH and the water absorbed by inorganics on phase state estimations**
RH is an important parameter affecting the phase state estimations. We perform a sensitivity calculation (case B, Table 1)
with the simulated RH increased by a factor of 10 % to compensate for the fact that the current model underestimates the
observed RH as shown in Fig. 2 and found in our previous simulations (Qu et al., 2019; Zhang et al., 2022). The increases in
simulated RH lead to more occurrence (the liquid frequency increased by 10 – 20 %) of liquid SOA particles in southeastern
China where the predicted RH is ranged mainly from 70 to 80 % (Fig. S4), with very limited effects in phase state
predictions in relatively dry areas, e.g., the northern China (Fig. 9a). Besides RH, the water absorbed by inorganic species
present in atmospheric particles also plays an important role in the phase state of ambient particles (Marcolli and Krieger,
2006). Here we perform a sensitivity calculation (case C, Table 1) assuming that the organic and inorganic compounds are
always internally mixed in one phase, which can be regarded that the predicted viscosity is at the lower end of the viscosity
in real ambient particles. Figure 9b shows that including the water absorbed by inorganic species can significantly lower the
simulated viscosity over most areas of China, with the liquid frequency increased by 15 – 45 % in the southeast, and 5 – 15 %
in some areas of the northeast. The effects over dry lands in the northwestern China are relatively small.
**4 Conclusions and discussions**
We simulate the glass transition temperature and viscosity of SOA particles over China based on SOA volatility using the
WRF-Chem model. This is the first time that spatial variations in the SOA phase state over China are investigated by a
regional chemical transport model. Simulations show that $T_g$ values of dry SOA ($T_{g,org}$) range from ~287 K to 305 K over
most areas of China at the surface, which is consistent with our previous simulated results based on a global transport model
(Shiraiwa et al., 2017). The $T_{g,org}$ is higher in the northwestern China than the southeastern China. This geospatial variation in
$T_{g,org}$ is related to the simulated SOA volatility distributions that SOA particles in northwestern China have relatively low
volatilities. Considering the water uptake by SOA particles, the SOA viscosity also shows a prominent geospatial gradient
that highly viscous or solid SOA particles are mainly found in the northwestern China. The frequency of liquid SOA particles



in most areas with the latitude higher than 30°N is less than 20 %. The lowest and highest SOA viscosities both occur over
the Qinghai-Tibet Plateau, with the solid phase state predicted over dry and high-altitude areas and the liquid phase state
predicted mainly in the south of the plateau with high RH in the summer. This indicates that this area has very large spatial
variation in SOA phase state, which was not found in our previous global simulations. We recommend measurements in
ambient particle phase state to be conducted over the Qinghai-Tibet Plateau, one of the most sensitive regions to climate
change (Liu and Chen, 2000).

The mixing timescale of organic molecules in 200 nm SOA particles is calculated based on the simulated particle

viscosity and the bulk diffusion coefficients of organic molecules. Calculations show that at the surface and at 500 hPa, the
percent time of $\tau_{mix}$ longer than 1 h is > ~ 70 % in the northwest of the "Hu Huanyong Line". When the mixing timescales of
organics are greater than roughly 1 h, which is longer than the typical time step in CTMs, the instantaneous equilibrium
partitioning usually assumed in SOA formation simulations is subject to be re-evaluated. The kinetic partitioning considering
the bulk diffusion in viscous particles may be required for more accurate prediction of SOA mass concentrations and size
distributions over the areas with long mixing timescale of organic molecules (Shiraiwa and Seinfeld, 2012; Zaveri et al.,
2018; Li and Shiraiwa, 2019; Zaveri et al., 2020; He et al., 2021; Maclean et al., 2021; Jathar et al., 2021; Shrivastava et al.,

2022).

The $T_{g,org}$ calculations in the base simulation are based on simulated volatility distributions of SOA formed from VOCs,

with five $C^*$ bins of 0.1, 1, 10, 100, and 1000 μg m$^{-3}$ at 298 K, with $\Delta H_{vap}$ of 142, 131, 120, 109, and 98 kJ mol$^{-1}$ used in the
five $C^*$ bins, respectively. The average $C^*$ and $T_{g,org}$ of the simulated SOA agree well with the values estimated from ambient
measurements of OOA volatilities at the IAP site in Beijing, where ELVOCs were not determined in the observed OOA (Xu
et al., 2019). We run a sensitivity simulation considering ELVOCs, with the lowest volatility bin having the $C^*$ of 0.0001 μg
m$^{-3}$ at 298 K and the $\Delta H_{vap}$ of 40 kJ mol$^{-1}$, following Knote et al. (2015). The sensitivity simulation shows that compared to
the base simulation, the frequency of a liquid phase state does not change in most areas of China, and incorporation of
ELVOCs mainly have a limited effect making SOA particles more viscous in some areas of the southeastern China. Our
previous study ever found that the $T_{g,org}$ of total OA in the SOAS (Southern Oxidant and Aerosol Study) campaign could
have a large variation (from 232 to 334 K) (Li et al., 2020) as the volatility distributions derived from different methods
could be different (Stark et al., 2017). The most credible predicted $T_{g,org}$ values were screened to be 313 – 330 K during the
SOAS campaign based on the most atmospherically relevant volatility distributions (Li et al., 2020). The volatility
distributions of organic aerosols derived from ambient volatility measurements are still limited over China (Xu et al., 2021;
Liu et al., 2021a; Feng et al., 2023). It needs more field volatility measurements to evaluate the effects of ELVOCs as well as
the enthalpy of vaporization on OA phase state estimations over China.

In the base simulation we assume that SOA components are phase separated from inorganic compounds in fine particles,

in which way the organic-rich and inorganic-rich phases may undergo phase transition separately (Dette and Koop, 2015).



This is consistent to recent ambient observations showing that the phase separation with an organic-rich shell and an
inorganic core was a frequent phenomenon in individual particles (diameters > 100 nm) collected over the North China Plain
(Li et al., 2021a). Previous simulations conducted for the SOAS campaign also showed that the phase separation was
expected ~ 65 % of the time on average (Schmedding et al., 2020), and exhibited a diurnal variation with a relation with the
predicted O:C ratio and RH (Pye et al., 2017). From a sensitivity simulation assuming that the organic and inorganic
compounds are always internally mixed in one phase, we show that the water absorbed by inorganic species has a significant
impact lowering the simulated viscosity over the southeastern China. At such mixing condition with one phase, on one hand,
it is expected that the inorganic salts that often have lower $T_g$ compared to SOA compounds would further lower the particle
viscosity relative to the organic fraction alone (Dette and Koop, 2015). On the other hand, the presence of divalent inorganic
ions could increase the viscosity of mixed organic-inorganic particles, enabling a humidity-dependent gel phase transition
through cooperative ion-molecule interactions (Richards et al., 2020). For complex mixtures of primary OA, SOA and
inorganics, it was found that three distinct phases could occur (Huang et al., 2021). Solid organic-coated ammonium sulfate
particles were observed at high RH in the summertime Arctic atmosphere, which may be formed from contact efflorescence
during collision of an Aitken mode sulfate particle with an organic-coated ammonium sulfate particle (Kirpes et al., 2022).
Kinetic simulations of the condensation of SVOCs into a core-shell phase-separated particle showed that the interplay of
non-ideality and phase state can impact SOA partitioning kinetics significantly (Schervish and Shiraiwa, 2023). The impacts
of the complex interplay of organic and inorganic compounds on the phase state of multicomponent particles in ambient air
warrant further investigations in future studies (Lilek and Zuend, 2022).
*Data availability.* The simulation data in this study are available upon request from the corresponding author
(liying-iap@mail.iap.ac.cn).
*Supplement.* The supplement related to this article is available on-line.
*Author contributions.* YL and MS designed the research. ZZ, HR, and YQ performed the WRF-Chem modeling. ZZ, YL,
and HR analyzed the simulation data. WZ, WX, WH, and YS provided observation data. YL, ZZ, and MS wrote the
manuscript. ZZ, YL, and HR wrote the supplement. All authors discussed the results and contributed to the article editing.
*Competing interests.* At least one of the (co-)authors is a member of the editorial board of Atmospheric Chemistry and
Physics. The peer-review process was guided by an independent editor, and the authors have also no other competing
interests to declare.
*Acknowledgements.* This work was supported by the National Natural Science Foundation of China (grant no. 42075110).
The authors thank Qi Chen at the Peking University and Bin Zhao at the Tsinghua University for insightful suggestions on



SOA volatility simulations in this work.
*Financial support.* This work was supported by the National Natural Science Foundation of China (grant no. 42075110).

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



**Table 1.** Sensitivity calculations for evaluating the effects of simulated SOA volatility distributions (sensitivity case A), RH (sensitivity
case B) and the water absorbed by inorganic components (sensitivity case C) on viscosity estimations.

| Cases | $C^*$ at 298 K and $\Delta H_{vap}$ in each volatility bin | The liquid water content considered in viscosity estimations | RH |
|---|---|---|---|
| Base case | 0.1, 1, 10, 100, and 1000 μg m$^{-3}$ with $\Delta H_{vap}$ of 142, 131, 120, 109, and 98 kJ mol$^{-1}$ | water absorbed by SOA particles with the assumption that SOA particles are externally mixed with inorganics | RH simulated by WRF-Chem |
| Sensitivity case A | 0.0001, 1, 10, 100, and 1000 μg m$^{-3}$ with $\Delta H_{vap}$ of 40, 131, 120, 109, and 98 kJ mol$^{-1}$ | water absorbed by SOA particles with the assumption that SOA particles are externally mixed with inorganics | RH simulated by WRF-Chem |
| Sensitivity case B | 0.1, 1, 10, 100, and 1000 μg m$^{-3}$ with $\Delta H_{vap}$ of 142, 131, 120, 109, and 98 kJ mol$^{-1}$ | water absorbed by SOA particles with the assumption that SOA particles are externally mixed with inorganics | RH simulated by WRF-Chem increased by a factor of 10% |
| Sensitivity case C | 0.1, 1, 10, 100, and 1000 μg m$^{-3}$ with $\Delta H_{vap}$ of 142, 131, 120, 109, and 98 kJ mol$^{-1}$ | water absorbed by both SOA particles and inorganic components with the assumption that SOA particles are internally mixed with inorganics | RH simulated by WRF-Chem |





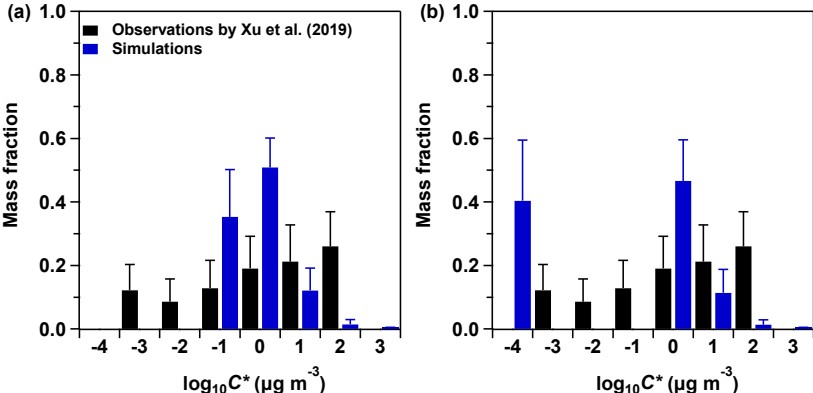

**Figure 1.** Comparison of the average volatility distributions of observed OOA and simulated SOA at the IAP site during 20 May – 23 June 2018. The black bars represent the volatility distributions of observed OOA adopted from Xu et al. (2019). The blue bars represent the volatility distributions of SOA simulated by WRF-Chem, with five $C^*$ bins set to be 0.1, 1, 10, 100, and 1000 μg m$^{-3}$ at 298 K in (a), and 0.0001, 1, 10, 100, and 1000 μg m$^{-3}$ at 298 K in (b). The blue error bars represent the one standard deviation.



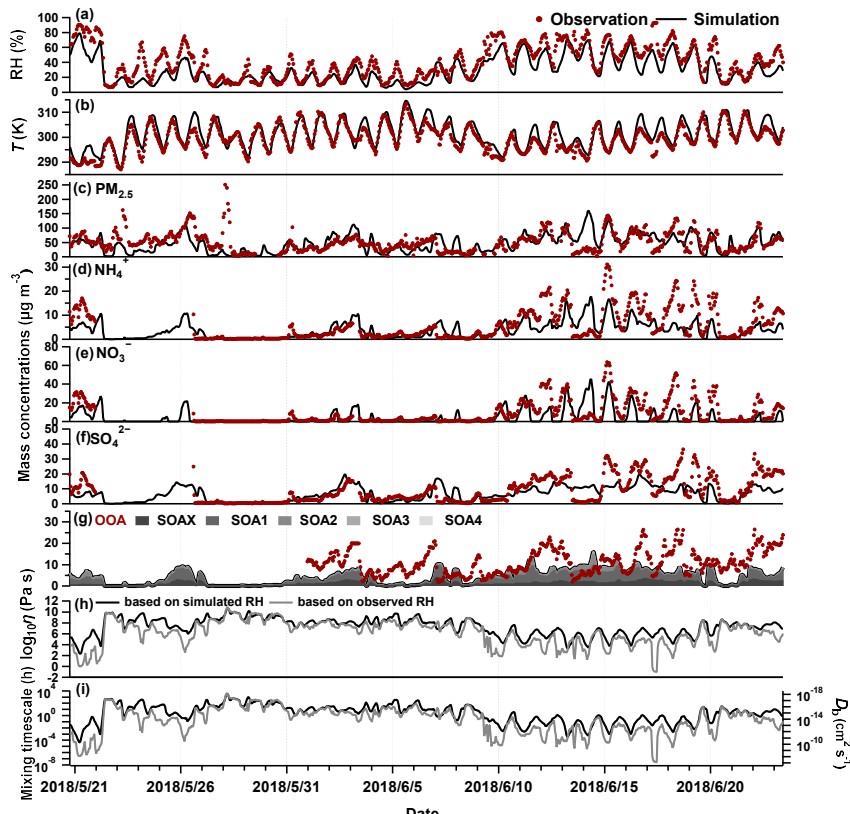

815

**Figure 2.** Observations and simulations of temporal variations of (a) RH, (b) $T$, (c) PM$_{2.5}$ concentrations, (d) NH$_4^+$ concentrations, (e) NO$_3^-$ concentrations, and (f) SO$_4^{2-}$ concentrations at the IAP site. (g) Observed OOA concentrations (red dots) and simulated SOA concentrations, with SOAX, SOA1, SOA2, SOA3, and SOA4 represent the SOA with $C^*$ of 0.1, 1, 10, 100, and 1000 μg m$^{-3}$ at 298 K, respectively. (h) SOA viscosity, and (i) bulk diffusion coefficients and mixing timescale of organic molecules within 200 nm SOA particles calculated using the RH simulated by the WRF-Chem model or the RH observed at the IAP site.










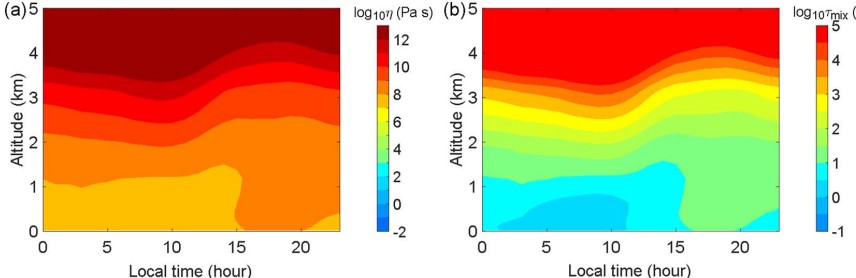

**Figure 3.** Median diurnal and vertical profiles of estimated (a) SOA viscosity and (b) mixing timescales for organic molecules within 200 nm SOA particles at the IAP site during May 20 – June 23 in 2018. Note: altitude is approximate and estimated from WRF pressure layers.



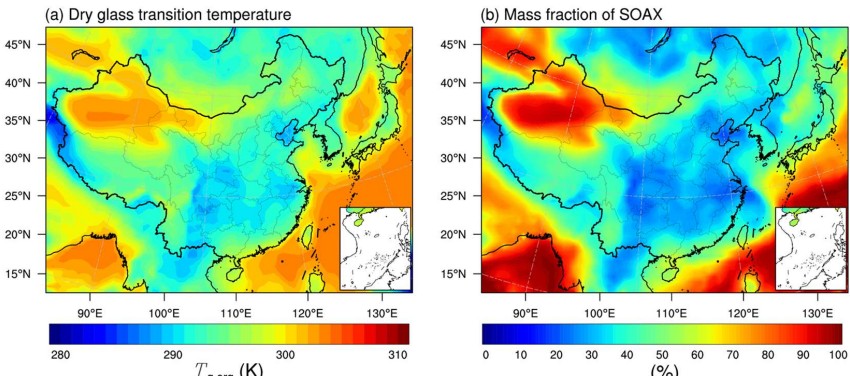

**Figure 4.** The predicted median surface values of (a) glass transition temperature of SOA particles at the dry condition and (b) mass fractions of SOAX ($C^* = 0.1$ µg m$^{-3}$ at 298 K) during May 20 – June 23 in 2018 simulated in the base case (Table 1).





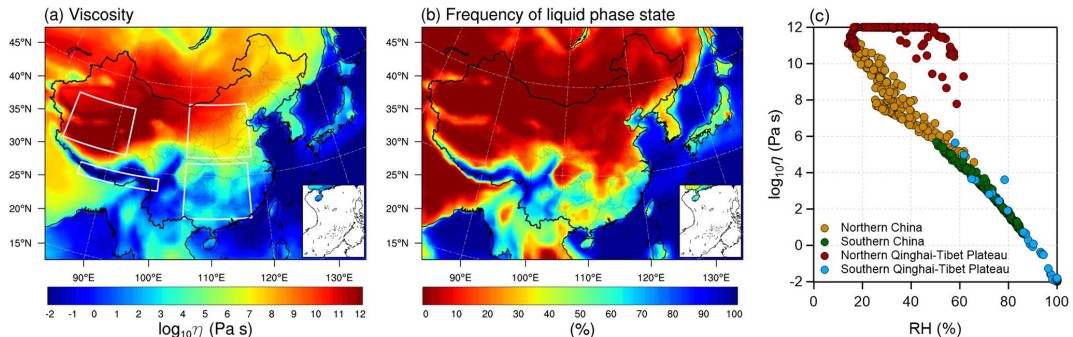


**Figure 5.** WRF-Chem predicted (a) median surface values of viscosity and (b) the percent time that an organic aerosol particle is in the
liquid phase state during May 20 – June 23 in 2018. (c) The median values of viscosity as a function of RH calculated for selected regions
in the northern China, southern China, northern Qinghai-Tibet Plateau, and southern Qinghai-Tibet Plateau as specified by white boxes in
panel (a).


















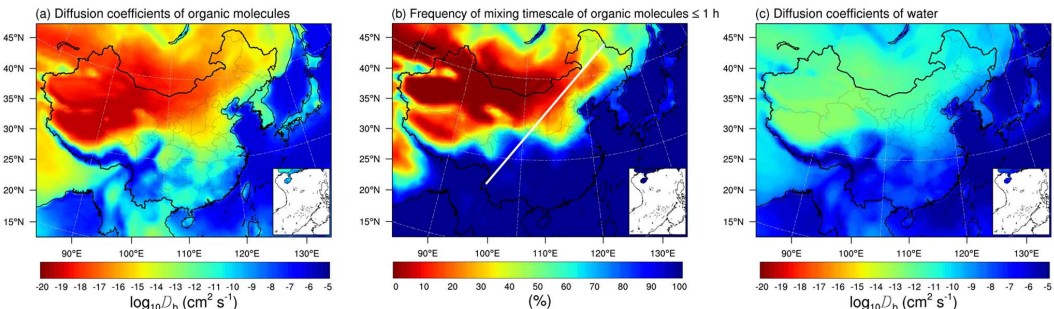

**Figure 6.** WRF-Chem predicted median surface values of the diffusion coefficients of (a) organic molecules and (c) water molecules in SOA particles. (b) The percent time that the mixing timescale of organic molecules in a 200 nm particle is less than 1 h during May 20 – June 23 in 2018. The white line indicates the "Hu Huanyong Line" (Hu, 1935).



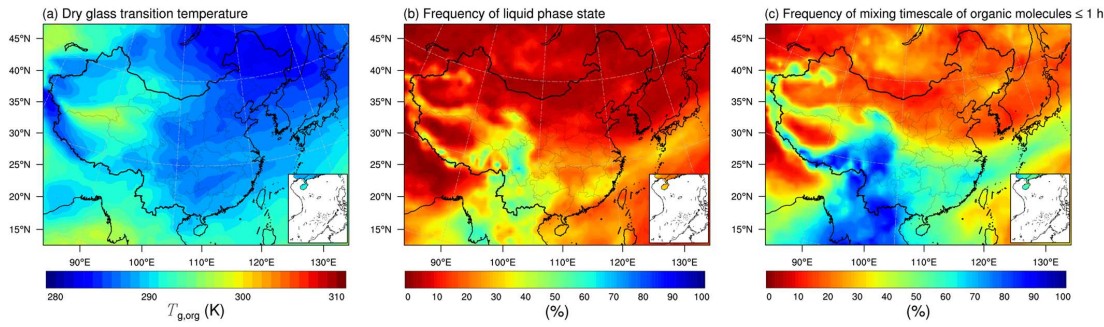

**Figure 7.** WRF-Chem predicted median values of (a) glass transition temperature of SOA particles at the dry condition, (b) the percent time that an organic aerosol particle is in the liquid phase state, and (c) the percent time that the mixing timescale of organic molecules in a 200 nm particle is less than 1 h at 500 hPa during May 20 – June 23 in 2018.



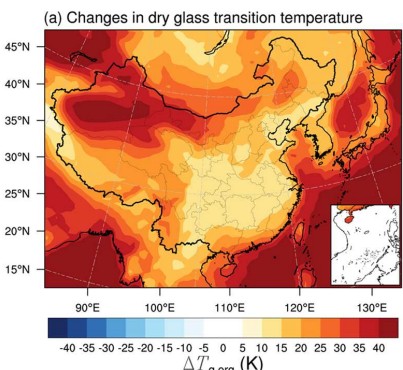
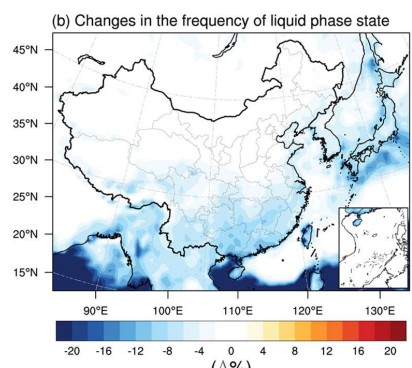

**Figure 8.** Modelled median differences of (a) glass transition temperature of SOA particles at the dry condition, and (b) the percent time that an organic aerosol particle is in the liquid phase state between a sensitivity case with the lowest $C^*$ of 0.0001 μg m$^{-3}$ at 298 K ($\Delta H_{vap}$ of 40 kJ mol$^{-1}$, case A in Table 1) and a base case with the lowest $C^*$ of 0.1 μg m$^{-3}$ at 298 K ($\Delta H_{vap}$ of 142 kJ mol$^{-1}$, Table 1).





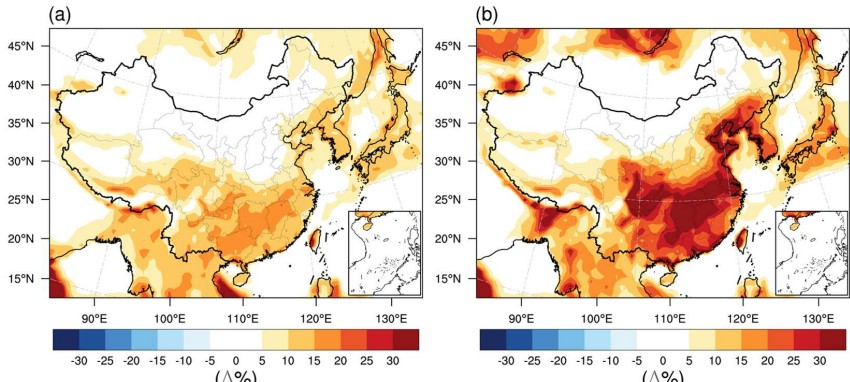

**Figure 9.** (a) Impacts of RH on the percent time that an organic aerosol particle is in the liquid phase state: modelled median differences between a case with the WRF-Chem simulated RH increased by a factor of 10 % (case B, Table 1) and a base case (Table 1) with the WRF-Chem simulated RH. (b) Impacts of the water absorbed by inorganics on the percent time that an organic aerosol particle is in the liquid phase state: modelled median differences between a case considering the water absorbed by both SOA particles and inorganics (case C, Table 1) and a base case (Table 1) considering the water absorbed by SOA particles solely.