# Peer review of "Phase state and viscosity of secondary organic aerosols over China simulated by WRF-Chem"

_EGUsphere, 2023_

## Author Comment (AC1)

**Response to the comments of Anonymous Referee #2**
**Referee General Comment:**

I am interested in the content of this paper and very concerned about the research results. From the perspective of scientific significance, I support the publication of this paper. However, reading this paper is very difficult, and I suggest adjusting the paper to highlight the main theme.

Response: We thank the positive review and very helpful suggestions from the Referee #2. We have carefully revised the manuscript as below.

**Referee Specific Comment:**
1. Firstly, I hope to see a clear conclusion. It feels like the paper ends in the midst of discussion. Please clarify the conclusion of the paper in the revised manuscript. If there is more to discuss, please use a separate subsection for that discussion.

Response: Thanks. We have inserted the discussion into the Result Section and revised the conclusion as follows.

Line 383 – 419: "We previously developed a new parameterization predicting the glass transition temperature of an organic compound as a function of its volatility (Li et al., 2020). Based on this new parameterization, we use the WRF-Chem model and simulate the $T_g$ and viscosity of SOA particles over China in June of 2018. This is the first time that spatial variations in the SOA phase state over China are investigated on a regional scale. The main conclusions are summarized below.

(1) Simulations show that $T_g$ values of dry SOA ($T_{g,org}$) range from ~287 K to 305 K over most areas of China at the surface, consistent with our previous simulated results based on a global transport model (Shiraiwa et al., 2017). The $T_{g,org}$ is higher in the northwestern China than the southeastern China. This geospatial variation in $T_{g,org}$ is related to the simulated SOA volatility distributions that SOA particles in northwestern China have relatively low volatilities.

(2) Considering the water uptake by SOA particles, the SOA viscosity also shows a prominent geospatial gradient that highly viscous or solid SOA particles are mainly found in the northwestern China. The frequency of liquid SOA particles in most areas with the latitude higher than 30 °N is less than 20 %. A very large spatial variation in SOA phase state over the Qinghai-Tibet Plateau was found and we recommend measurements in ambient particle phase state to be conducted over this area, one of the most sensitive regions to climate change.

(3) The mixing timescale of organic molecules in 200 nm SOA particles is calculated based on the simulated particle viscosity and the bulk diffusion coefficients of organic molecules. Calculations show that at the surface and at 500 hPa, the percent time of $\tau_{mix}$ longer than 1 h is > ~ 70 % in the northwest of the "Hu Huanyong Line". The implication of this result is that when the $\tau_{mix}$ values are greater than roughly 1 h, which is longer than the typical time step in CTMs, the instantaneous equilibrium

partitioning usually assumed in SOA formation simulations is subject to be re-evaluated. We recommend to test the effects of kinetic partitioning considering the bulk diffusion in viscous particles on the prediction of SOA mass concentrations and size distributions over the areas with long mixing timescale of organic molecules.

(4) The average volatility ($C^*$) and $T_{g,org}$ of the simulated SOA agree well with the values estimated from ambient measurements of OOA volatilities at the IAP site in Beijing, where ELVOCs were not determined in the observed OOA (Xu et al., 2019). The sensitivity simulation considering the formation of ELVOCs shows that compared to the base simulation, the frequency of a liquid phase state does not change in most areas of the northern China. In some areas of the southeastern China the SOA particles become more viscous with the percent time that a SOA particle is in the liquid phase state decreases by up to 12 %. It needs more field volatility measurements to evaluate the effects of ELVOCs as well as the enthalpy of vaporization on OA phase state estimations over China.

(5) Differed from the base simulation that SOA components are assumed to be phase separated from inorganic compounds in particles, we conduct a sensitivity simulation assuming that the organic and inorganic compounds are internally mixed in one phase. We show that the water absorbed by inorganic species has a significant impact lowering the simulated viscosity over the southeastern China, with the liquid frequency increased by 15 – 45 %. Future work should consider the effects of the mixing state of organic and inorganic compounds on the simulations of phase state of multicomponent particles in ambient air.

In summary, our simulations demonstrate the spatial distributions of the glass transition temperature and viscosity of SOA particles over China on a regional scale for the first time. The further calculations of the mixing timescale of organic molecules in SOA particles have an implication of the need to evaluate the effects of the phase state on predictions of SOA gas-particle partitioning, and thus the SOA mass concentrations and size distributions in CTMs."

2. The theme of the paper is not sufficiently clear, and there is a discrepancy between the focus of the conclusions in the conclusion section and those in the abstract. It is unclear what your main focus is. Please clarify your focus and then revise the paper accordingly, making choices about what to include. For instance, you might consider relocating some figures from the main body of the paper to the appendix, and vice versa.

Response: In response to this comment, we have re-written the conclusion section (see the response to your comment 1) making it more consistent with the focus of the abstract. As the glass transition temperature, the viscosity of SOA particles, the bulk diffusion coefficients, and the mixing timescale in SOA particles have not been shown in previous studies on a regional scale, we kept all the figures in the main text file for comparison by future studies. We have revised the manuscript to make the structure more clear and the main theme highlighted. The revision is included below.

Line 26 – 27: "We also calculate the characteristic mixing timescale of organic molecules in 200 nm SOA particles to evaluate kinetic limitations in SOA partitioning."

Line 59 – 61: "Accurate predictions of the viscosity need the information of molecular structures and functional groups (Song et al., 2016; Rothfuss and Petters, 2017; Gervasi et al., 2020; Galeazzo and Shiraiwa, 2022); however, molecular specificity is often unavailable in ambient measurements, leading to the prediction of the phase state of ambient SOA particles difficult."

Line 82 – 84: "It is needed to conduct simulations to investigate the SOA phase state varied with locations and the time. Simulations of the SOA phase state in China on a regional scale have not been available."

Line 232 – 239: "We highlight the mixing timescale of 1 hour as the time step adopted in CTMs is often ~ 0.5 – 1 hour (Maclean et al., 2021). Current CTMs usually assume that the gas-particle partitioning of SVOCs reaches equilibrium quickly within the time step (Pankow, 1994; Donahue et al., 2006). When the mixing timescales of organics within SOA particles are $\leqslant$ ~ 1 hour, the instantaneous equilibrium is a reasonable assumption. However, when the mixing timescales of organics are longer than ~ 1 hour, non-equilibrium between the gas phase and the particle phase, i.e., the kinetic partitioning may need to be considered in simulating the SOA formation in CTMs (Shiraiwa and Seinfeld, 2012; Zaveri et al., 2018; Li and Shiraiwa, 2019; Zaveri et al., 2020; He et al., 2021; Jathar et al., 2021; Maclean et al., 2021; Shiraiwa and Pöschl, 2021; Shrivastava et al., 2022)."

Line 288 – 291: "The phase state of SOA particles is affected by ambient conditions and the particle chemical composition (Koop et al., 2011). Figure 6 shows the median values of viscosity as a function of RH, $T$ and the mass fraction of low-volatility compound (SOAX with $C^*$ of 0.1 μg m$^{-3}$ at 298 K) calculated for selected regions in the northern China, southern China, northern Qinghai-Tibet Plateau, and southern Qinghai-Tibet Plateau as specified by white boxes in Fig. 5a."

Line 365 – 381: "Besides RH, the mixing state of the organic and inorganic species in atmospheric particles also plays an important role in the phase state of ambient particles. The SOA components are assumed to be phase separated from inorganic compounds in particles in our base simulation, which is consistent to recent ambient observations showing that the phase separation with an organic-rich shell and an inorganic core was a frequent phenomenon in individual particles (diameters > 100 nm) collected over the North China Plain (Li et al., 2021a). To assess the potential effects of inorganic compounds on the phase state of ambient particles, we perform a sensitivity calculation (case C, Table 1) assuming that the organic and inorganic compounds are internally mixed in one phase. In this case the water absorbed by inorganic species can lower the particle viscosity relative to the organic fraction alone. Figure 10b shows that the water associated with inorganic species can significantly

lower the viscosity over most areas of China, with the liquid frequency increased by 15 – 45 % in the southeast, and 5 – 15 % in some areas of the northeast. The effects over dry lands in the northwestern China are relatively small. Previous studies showed that at such mixing condition with one phase, on one hand, it is expected that the inorganic salts that often have lower $T_g$ compared to SOA compounds would further lower the particle viscosity (Dette and Koop, 2015). On the other hand, the presence of divalent inorganic ions could increase the viscosity of mixed organic-inorganic particles, enabling a humidity-dependent gel phase transition through cooperative ion-molecule interactions (Richards et al., 2020). For complex mixtures of primary OA, SOA and inorganics, it was found that three distinct phases could occur (Huang et al., 2021). The impacts of the mixing state of organic and inorganic compounds on the phase state of multicomponent particles in ambient air warrant further investigations in future studies (Lilek and Zuend, 2022; Schervish and Shiraiwa, 2023)."

We have moved a scatter plot in the original supplement file to the main text file as below. This new figure shows that the factors including RH, $T$ and the SOA volatility affect the viscosity predictions.

[Figure]

**Figure 6.** The median values of viscosity as a function of (a) RH, (b) $T$ and (c) the mass fraction of SOAX ($C^* = 0.1$ µg m$^{-3}$ at 298 K) calculated for selected regions in the northern China, southern China, northern Qinghai-Tibet Plateau, and southern Qinghai-Tibet Plateau as specified by white boxes in Fig. 5a during May 20 – June 23 in 2018.

---

## Author Comment (AC2)

**Response to the comments of Anonymous Referee #1**

**Referee General Comments:**

1. The authors claim that they are the first to investigate the spatial distributions of the SOA phase state over China by a regional CTM. But what about the model being applied to other parts of the world and have the authors looked into those applications and how does it differ from application of the regional CTM in those areas as compared to China?

Response: We appreciate very much that the Anonymous Referee #1 takes time looking into our manuscript carefully and gives the positive review and very helpful suggestions.

Simulation of the phase state of ambient SOA particles using CTMs is a relatively new topic and the previous studies are only a few. In the Introduction section of our submitted manuscript (Line 68 – 76), we included previous studies including Schmedding et al. (2020) and Li et al. (2021b) focusing on the U.S. These previous studies stemmed from our global SOA phase state simulation (Shiraiwa et al., 2017) which is the first developing a method which can successfully predict the phase state of ambient SOA particles.

Later we developed another parameterization predicting the viscosity as a function of volatility (Li et al., 2020). Then Rasool et al. (2021), Shrivastava et al. (2022), and Rasool et al. (2023) applied this new parameterization simulating the SOA phase state over the Amazon rainforest. Therefore, those previous studies used different parameterizations and the viscosity prediction method developed by us simulating the SOA phase state. The current study also applied the parameterization in Li et al. (2020). We compared the current simulations over China with our previous global simulations (Shiraiwa et al., 2017). In the revised manuscript we further modified the previous applications of regional CTMs over other locations of the world as follows.

Line 59 – 61: "Accurate predictions of the viscosity need the information of molecular structures and functional groups (Song et al., 2016; Rothfuss and Petters, 2017; Gervasi et al., 2020; Galeazzo and Shiraiwa, 2022); however, molecular specificity is often unavailable in ambient measurements, leading to the prediction of the phase state of ambient SOA particles difficult."

Line 63 – 79: "Li, Shiraiwa and coauthors first developed a parameterization predicting the glass transition temperature ($T_g$) based on the molar mass ($M$) and the atomic O/C ratio for carbon-hydrogen (CH) and carbon-hydrogen-oxygen (CHO) compounds with their molar mass less than 450 g mol$^{-1}$ (Shiraiwa et al., 2017)……. This parameterization has been successfully coupled into CTMs simulating the SOA phase state over the globe (Shiraiwa et al., 2017) or the U.S. (Schmedding et al., 2020; Li et al., 2021b), showing that semi-solid and amorphous solid phase states frequently occurred over dry lands and in the upper troposphere. Further parameterizations were developed to predict $T_g$ as a function of the saturation mass concentration ($C^0$) and the O/C ratio of organic compounds, or as a function of $C^0$ solely, which indirectly included the effect of molecular structure on $T_g$ estimations (Li et al., 2020). This

parameterization can be used in the volatility basis set (VBS) framework (Donahue et al., 2006), which is widely adopted in CMTs simulating SOA formation (Lane et al., 2008a; Knote et al., 2015). Rasool et al. (2021) then coupled the new method (Li et al., 2020) into the Weather Research and Forecasting Model coupled to chemistry (WRF-Chem) (Grell et al., 2005; Fast et al., 2006), and the simulations showed that the viscosity of SOA particles could be reasonably predicted during the dry-to-wet transition season in the Amazon rainforest. Li et al. (2020) was also applied in the WRF-Chem simulating the effects of particle phase state on the multiphase chemistry of SOA formation in the Amazon rainforest (Shrivastava et al., 2022; Rasool et al., 2023)."

Line 82 – 84: "It is needed to conduct more simulations to investigate the SOA phase state varied with locations and the time. Simulations of the SOA phase state in China on a regional scale have not been available."

2. I don't know if I am missing something, but I could not find Fig. S1, S2, S3, S4, S5, S6, S7 and Table S1, S2, which are referred to throughout the manuscript.

Response: They were contained in the Supplement file. In the revised manuscript we have added "in the Supplement" after referring to the figures and tables in the Supplement which can be found in the following link:

https://egusphere.copernicus.org/preprints/2023/egusphere-2023-1444/egusphere-2023-1444-supplement.pdf

3. In general the figures are mostly contour plots of different parameters. A more analytical presentation of the results from the model is missing in the current form of the manuscript. The authors may try to add some statistical figures or tables in order to analyze the outcomes from the model.

Response: Thanks for this very helpful suggestion. In our original submitted manuscript we already included a table summarizing the model performance statistics (Table S2 in the Supplement) for simulated meteorological parameters and the concentrations of $PM_{2.5}$ and $PM_{2.5}$ components. We did not give a statistical figure for the simulated SOA viscosity etc as the observations of the viscosity of ambient particles are few; thus we compared the predictions of the phase state (i.e., liquid, semi-solid or amorphous solid) with particle bounce measurements. Considering your suggestion as well as the comment 2 of Referee #2, we have moved a scatter plot in the original supplement file to the main text file which clearly shows that the viscosity is very different for different regions. This new figure also shows that the factors including RH, $T$ and the SOA volatility affect the viscosity predictions.

[Figure]

**Figure 6.** The median values of viscosity as a function of (a) RH, (b) $T$ and (c) the mass fraction of SOAX ($C^* = 0.1$ µg m$^{-3}$ at 298 K) calculated for selected regions in the northern China, southern China, northern Qinghai-Tibet Plateau, and southern Qinghai-Tibet Plateau as specified by white boxes in Fig. 5a during May 20 – June 23 in 2018.

4. The conclusion section seems to be lengthy. May be the authors can think about making a separate discussions section and the conclusion section.

Response: Thanks for the suggestion. We have inserted the discussion into the Result Section and revised the conclusion as follows.

Line 383 – 419: "We previously developed a parameterization predicting the glass transition temperature of an organic compound as a function of its volatility (Li et al., 2020). Based on this new parameterization, we use the WRF-Chem model and simulate the $T_g$ and viscosity of SOA particles over China in June of 2018. This is the first time that spatial variations in the SOA phase state over China are investigated on a regional scale. The main conclusions are summarized below.

(1) Simulations show that $T_g$ values of dry SOA ($T_{g,org}$) range from ~287 K to 305 K over most areas of China at the surface, consistent with our previous simulated results based on a global transport model (Shiraiwa et al., 2017). The $T_{g,org}$ is higher in the northwestern China than the southeastern China. This geospatial variation in $T_{g,org}$ is related to the simulated SOA volatility distributions that SOA particles in northwestern China have relatively low volatilities.

(2) Considering the water uptake by SOA particles, the SOA viscosity also shows a prominent geospatial gradient that highly viscous or solid SOA particles are mainly found in the northwestern China. The frequency of liquid SOA particles in most areas with the latitude higher than 30°N is less than 20 %. A very large spatial variation in SOA phase state over the Qinghai-Tibet Plateau was found and we recommend measurements in ambient particle phase state to be conducted over this area, one of the most sensitive regions to climate change.

(3) The mixing timescale of organic molecules in 200 nm SOA particles is calculated based on the simulated particle viscosity and the bulk diffusion coefficients of organic molecules. Calculations show that at the surface and at 500 hPa, the percent time of $\tau_{\text{mix}}$ longer than 1 h is $> \sim 70$ % in the northwest of the "Hu Huanyong Line". The implication of this result is that when the $\tau_{\text{mix}}$ values are greater than roughly 1 h, which is longer than the typical time step in CTMs, the instantaneous equilibrium partitioning usually assumed in SOA formation simulations is subject to be re-evaluated. We recommend to test the effects of kinetic partitioning considering the bulk diffusion in viscous particles on the prediction of SOA mass concentrations and size distributions over the areas with long mixing timescale of organic molecules.

(4) The average volatility ($C^*$) and $T_{\text{g,org}}$ of the simulated SOA agree well with the values estimated from ambient measurements of OOA volatilities at the IAP site in Beijing, where ELVOCs were not determined in the observed OOA (Xu et al., 2019). The sensitivity simulation considering the formation of ELVOCs shows that compared to the base simulation, the frequency of a liquid phase state does not change in most areas of the northern China. In some areas of the southeastern China the SOA particles become more viscous with the percent time that a SOA particle is in the liquid phase state decreases by up to 12 %. It needs more field volatility measurements to evaluate the effects of ELVOCs as well as the enthalpy of vaporization on OA phase state estimations over China.

(5) Differed from the base simulation that SOA components are assumed to be phase separated from inorganic compounds in particles, we conduct a sensitivity simulation assuming that the organic and inorganic compounds are internally mixed in one phase. We show that the water absorbed by inorganic species has a significant impact lowering the simulated viscosity over the southeastern China, with the liquid frequency increased by 15 – 45 %. Future work should consider the effects of the mixing state of organic and inorganic compounds on the simulations of phase state of multicomponent particles in ambient air.

  In summary, our simulations demonstrate the spatial distributions of the glass transition temperature and viscosity of SOA particles over China on a regional scale for the first time. The further calculations of the mixing timescale of organic molecules in SOA particles have an implication of the need to evaluate the effects of the phase state on predictions of SOA gas-particle partitioning, and thus the SOA mass concentrations and size distributions in CTMs."

**Referee Specific Comments:**
5. Page 3 Line 63: What is "semi-sold"?

Response: Thanks for spotting this typo. We have changed it to "semi-solid" in the revised manuscript.

6. Page 3 Line 76: Reference missing year "Maclean et al.".

Response: We have added the year in the revised manuscript.

7. Page 4 Line 107:   What is the "outer domain" considered here?

Response: Thanks. In the revised manuscript we have moved the sentence that "We set up two domains (Fig. S1 in the Supplement) with the horizontal resolutions of 81 km and 27 km, respectively, and 18 vertical layers are applied from the surface up to 100 hPa" to a place (Line 110) before introducing the "outer domain" which indicates the simulation domain 1 in the Fig. S1 in the Supplement.

8. Page 4 line 115: "39 nm to 10 μm". it would be nice to use any one of units, either nm or μm.

Response: It has been modified to "0.039 μm to 10 μm" in the revised manuscript.

9. Page 9 Line 271 and Line 274: What does the author mean here by "Our previous global simulations"? Please provide proper reference.

Response: It is the Shiraiwa et al. (2017), the first study giving a global picture of the phase state of SOA particles, which was added as a reference at Line 272 in the original manuscript. We also have added it at Line 283 at the end of the sentence in the revised manuscript.

10. Page 10 Line 295: "(see the Method)". What "method" is referred to here?

Response: It refers to the Method section in our manuscript. We clarified this in the revised manuscript. The bulk diffusivity is firstly introduced in the Introduction section (Line 48 − 49), and its calculation method can be found in the Method section (Line 155 − 157) and the Supplement.

11. Page 10 Line 297: "The highest value is ~$10^{-5}$ cm$^2$ s$^{-1}$ occurring in liquid SOA particles in the southern Tibetan Plateau (Fig. 6a).". Why is the highest value being observed at the southern Tibetan Plateau? Can the authors provide a probable reasoning for this observation?

Response: This is because the SOA particles in the southern Tibetan Plateau have the lowest viscosity values compared to other regions of China. The viscosity is inversely related to the $D_b$ according to the Stokes-Einstein relation. We have modified the sentence as below:

"The highest value is ~$10^{-5}$ cm$^2$ s$^{-1}$ occurring in the SOA particles in the southern Tibetan Plateau (Fig. 7a) because of the very low viscosity simulated over this region (Fig. 5a)".

12. Figure 2: Figure 2g is a bit difficult to read. Maybe the authors can try having a different color scale for SOAX, SOA1, SOA2, SOA3 and SOA4.

Response: We used new colors following your suggestion as shown below. SOA3 and SOA4 are not very obvious because they have relatively high volatility leading to their mass concentrations in the particle phase very low.

[Figure]

**Figure 2.** Observations and simulations of temporal variations of (a) RH, (b) *T*, (c) PM$_{2.5}$ concentrations, (d) NH$_4^+$ concentrations, (e) NO$_3^-$ concentrations, and (f) SO$_4^{2-}$ concentrations at the IAP site. (g) Observed OOA concentrations (red dots) and simulated SOA concentrations, with SOAX, SOA1, SOA2, SOA3, and SOA4 represent the SOA with *C*$^*$ of 0.1, 1, 10, 100, and 1000 μg m$^{-3}$ at 298 K, respectively. (h) SOA viscosity, and (i) bulk diffusion coefficients and mixing timescale of organic molecules within 200 nm SOA particles calculated using the RH simulated by the WRF-Chem model or the RH observed at the IAP site.

13. Figure 3: The vertical profiles of mixing timescales for organic molecules within 200 nm SOA particles at IAP site as shown in Figure 3b seems to be more sensitive to altitude than the SOA viscosity which is shown in Figure 3a. Can the authors provide a reasoning for this behavior?

Response: Thanks. This is related to the different scales and resolution of the two color bars used in Fig. 3a and 3b. We have replotted it in the revised manuscript as shown below.

[Figure]

**Figure 3.** Median diurnal and vertical profiles of estimated (a) SOA viscosity and (b) mixing timescales for organic molecules within 200 nm SOA particles at the IAP site during May 20 – June 23 in 2018. Note: altitude is approximate and estimated from WRF pressure layers.

14. The color bar used in figure 8a from -45 to 45 K. But in the contour plot I can only see the values above 0 or -5. Is it possible to make these color bars a bit more according to the range of values so as to see the variation more clearly? Same for Figure 8b and Figure 9.

Response: Thanks for this helpful suggestion. We included both the positive and negative values in the color bars because we wanted the readers to intuitively see whether the trend was increasing (with the warm color) or decreasing (with the cold color) in the sensitivity simulations. We appreciate you would like to allow us keeping the color bars as they were to show how the predicted SOA phase state changes with the volatility distributions (Fig. 8) and the aerosol liquid water etc (Fig. 9). In Fig. 9 we used the range of -35 % to 35 % in the color bars to show that the inclusion of the water absorbed by inorganics plays a more important role than the uncertainty in simulated RH affecting the phase state prediction.

**References**

Rasool, Q. Z., Shrivastava, M., Liu, Y., Gaudet, B., and Zhao, B.: Modeling the Impact of the Organic Aerosol Phase State on Multiphase OH Reactive Uptake Kinetics and the Resultant Heterogeneous Oxidation Timescale of Organic Aerosol in the Amazon Rainforest, ACS Earth and Space Chemistry, https://doi.org/10.1021/acsearthspacechem.2c00366, 2023.

Shrivastava, M., Rasool, Q. Z., Zhao, B., Octaviani, M., Zaveri, R. A., Zelenyuk, A., Gaudet, B., Liu, Y., Shilling, J. E., Schneider, J., Schulz, C., Zöger, M., Martin, S. T., Ye, J., Guenther, A., Souza, R. F., Wendisch, M., and Pöschl, U.: Tight coupling of surface and in-plant biochemistry and convection governs key fine particulate components over the Amazon Rainforest, ACS Earth Space Chem., 6, 380-390, https://doi.org/10.1021/acsearthspacechem.1c00356, 2022.

---

## Author Response (AR2)

**Response to the comments of Anonymous Referee #2**
**Referee Specific Comment:**
1. The authors emphasized several times that "This is the first time that spatial distributions of the SOA phase state over China are investigated by a regional CTM." However, I don't feel this add any merits to this study. Is it because it's hard to apply such parameterizations in regional CTM, or people think it's meaningless to investigate this issue with regional CTM? The true value of the study should be stated here.

Response: We thank the positive review and further very helpful suggestions from the Referee #2. The phase state of SOA particles plays an important role in many aerosol related processes, e.g., gas-particle partitioning, heterogeneous and multiphase reactions, particle size distribution evolution and ice nucleation pathways. However, current CTMs usually assume that SOA particles are homogeneous and well-mixed liquids, thus rapid establishment of gas–particle equilibrium is often applied in simulations of SOA formation and partitioning.

Recent measurements show that SOA can exist in liquid, semi-solid or solid states. The semi-solid or solid phase state can prolong the equilibration timescales in the gas-particle partitioning, indicating a need of considering kinetic limitations in simulating the SOA partitioning into highly viscous particles. Therefore, it is important to know the spatial distributions of the SOA phase state and viscosity, the information of which is usually missing in current CTMs. The above aspects have been clearly stated in the Introduction section. In the revised abstract, we have deleted the sentence that "This is the first time that spatial distributions of the SOA phase state over China are investigated by a regional CTM". We revised the first sentence of the abstract as below to show the significance of this study. We also adjusted the Conclusion Section (refer to the reply to your comment 3).

Line 14 – 19: "Chemical transport models (CTMs), however, usually assume that SOA particles are homogeneous and well-mixed liquids, with rapid establishment of gas–particle equilibrium for simulations of SOA formation and partitioning. Missing the information of SOA phase state and viscosity in CTMs impedes accurate representation of SOA formation and evolution, affecting the predictions of aerosol effects on air quality and climate".

2. Also, I didn't understand why the authors choose summer 2018 as the simulation period in the abstract. In the main text, I can see that it's for comparison with observations. If so, the observation-simulation comparisons should be a major point, as should be highlighted in the abstract.

Response: We add the following sentence in the revised abstract to highlight the result of the observation-simulation comparison.

Line 23 – 24: "The simulated $T_g$ of dry SOA ($T_{g,org}$) agrees well with the value estimated from ambient volatility measurements at an urban site in Beijing".

3. I feel the implications are not fully investigated. After reading the whole article, it's still not clear why one should care so much about the viscosity, and how the improved simulation of viscosity would refresh our understanding on SOA and its atmospheric processes. These points should be more clearly stated and described.

Response: Please refer to the reply to your comment 1 that the importance of the phase state and viscosity in many aerosol related processes has been clearly stated in the Introduction section (particularly paragraph 2 of the Introduction). In the main text of the manuscript, we also calculated the diffusion coefficients and mixing timescales of organic molecules within SOA, which has implications in how to properly treat the SOA partitioning (instantaneous equilibrium vs. kinetic partitioning) in CTMs. For example, Line 242 – 248: "We highlight the mixing timescale of 1 hour as the time step adopted in CTMs is often ~ 0.5 – 1 hour (Maclean et al., 2021). Current CTMs usually assume that the gas-particle partitioning of SVOCs reaches equilibrium quickly within the time step (Pankow, 1994; Donahue et al., 2006). When the mixing timescales of organics within SOA particles are ⩽ ~ 1 hour, the instantaneous equilibrium is a reasonable assumption. However, when the mixing timescales of organics are longer than ~ 1 hour, non-equilibrium between the gas phase and the particle phase, i.e., the kinetic partitioning may need to be considered in simulating the SOA formation in CTMs (Shiraiwa and Seinfeld, 2012; Zaveri et al., 2018; Li and Shiraiwa, 2019; Zaveri et al., 2020; He et al., 2021; Jathar et al., 2021; Maclean et al., 2021; Shiraiwa and Pöschl, 2021; Shrivastava et al., 2022)."

Following your suggestion, we revised the abstract moving the sentence with implications of the mixing timescale that is calculated from SOA viscosity to the last sentence of the abstract, to highlight the implication of this study.

Line 40 – 45: "We also calculate the characteristic mixing timescale of organic molecules in 200 nm SOA particles to evaluate kinetic limitations in SOA partitioning. Calculations show that during the simulated period the percent time of the mixing timescale longer than 1 h is > 70 % at the surface and at 500 hPa in most areas of the northern China, indicating that kinetic partitioning considering the bulk diffusion in viscous particles may be required for more accurate prediction of SOA mass concentrations and size distributions over these areas".

We also adjusted the Conclusion Section deleting the sentence that "In summary, our simulations demonstrate the spatial distributions of the glass transition temperature and viscosity of SOA particles over China on a regional scale for the first time". We moved the implication of the mixing timescales to be the last paragraph:

Line 425 – 431: "(5) The mixing timescale of organic molecules in 200 nm SOA particles is calculated based on the simulated particle viscosity and the bulk diffusion coefficients of organic molecules. Calculations show that at the surface and at 500 hPa, the percent time of $\tau_{\mathrm{mix}}$ longer than 1 h is > ~ 70 % in the northwest of the "Hu

Huanyong Line". The implication of this result is that when the $\tau_{\mathrm{mix}}$ values are greater than roughly 1 h, which is longer than the typical time step in CTMs, the instantaneous equilibrium partitioning usually assumed in SOA formation simulations is subject to be re-evaluated. We recommend to test the effects of kinetic partitioning considering the bulk diffusion in viscous particles on the prediction of SOA mass concentrations and size distributions over the areas with long mixing timescale of organic molecules.